# A Multi-Parametric Approach for Characterising Cerebral Haemodynamics in Acute Ischaemic and Haemorrhagic Stroke

**DOI:** 10.3390/healthcare12100966

**Published:** 2024-05-08

**Authors:** Abdulaziz Alshehri, Ronney B. Panerai, Angela Salinet, Man Yee Lam, Osian Llwyd, Thompson G. Robinson, Jatinder S. Minhas

**Affiliations:** 1Cerebral Haemodynamics in Ageing and Stroke Medicine (CHiASM) Research Group, Department of Cardiovascular Sciences, University of Leicester, Leicester LE1 7RH, UK; aaha7@leicester.ac.uk (A.A.); rp9@le.ac.uk (R.B.P.); salinetangela@gmail.com (A.S.); man.lam@uhl-tr.nhs.uk (M.Y.L.); tgr2@leicester.ac.uk (T.G.R.); 2College of Applied Medical Sciences, University of Najran, Najran P.O. Box 1988, Saudi Arabia; 3NIHR Leicester Biomedical Research Centre, British Heart Foundation Cardiovascular Research Centre, Glenfield Hospital, Leicester LE3 9QP, UK; 4Wolfson Centre for Prevention of Stroke and Dementia, Department of Clinical Neurosciences, University of Oxford, Oxford OX1 2JD, UK; osian.llwyd@ndcn.ox.ac.uk

**Keywords:** cerebral autoregulation, stroke, baroreflex sensitivity, blood-flow velocity, transcranial doppler sonography

## Abstract

Background and Purpose: Early differentiation between acute ischaemic (AIS) and haemorrhagic stroke (ICH), based on cerebral and peripheral hemodynamic parameters, would be advantageous to allow for pre-hospital interventions. In this preliminary study, we explored the potential of multiple parameters, including dynamic cerebral autoregulation, for phenotyping and differentiating each stroke sub-type. Methods: Eighty patients were included with clinical stroke syndromes confirmed by computed tomography within 48 h of symptom onset. Continuous recordings of bilateral cerebral blood velocity (transcranial Doppler ultrasound), end-tidal CO_2_ (capnography), electrocardiogram (ECG), and arterial blood pressure (ABP, Finometer) were used to derive 67 cerebral and peripheral parameters. Results: A total of 68 patients with AIS (mean age 66.8 ± SD 12.4 years) and 12 patients with ICH (67.8 ± 16.2 years) were included. The median ± SD NIHSS of the cohort was 5 ± 4.6. Statistically significant differences between AIS and ICH were observed for (i) an autoregulation index (ARI) that was higher in the unaffected hemisphere (UH) for ICH compared to AIS (5.9 ± 1.7 vs. 4.9 ± 1.8 *p* = 0.07); (ii) coherence function for both hemispheres in different frequency bands (AH, *p* < 0.01; UH *p* < 0.02); (iii) a baroreceptor sensitivity (BRS) for the low-frequency (LF) bands that was higher for AIS (6.7 ± 4.2 vs. 4.10 ± 2.13 ms/mmHg, *p* = 0.04) compared to ICH, and that the mean gain of the BRS in the LF range was higher in the AIS than in the ICH (5.8 ± 5.3 vs. 2.7 ± 1.8 ms/mmHg, *p* = 0.0005); (iv) Systolic and diastolic velocities of the affected hemisphere (AH) that were significantly higher in ICH than in AIS (82.5 ± 28.09 vs. 61.9 ± 18.9 cm/s), systolic velocity (*p* = 0.002), and diastolic velocity (*p* = 0.05). Conclusion: Further multivariate modelling might improve the ability of multiple parameters to discriminate between AIS and ICH and warrants future prospective studies of ultra-early classification (<4 h post symptom onset) of stroke sub-types.

## 1. Introduction

Stroke is a life-threatening condition that affects millions of people globally. In 2016, strokes affected 13.7-million adults worldwide and resulted in 5.5-million death incidents, and in the United Kingdom, there were 134,979 stroke cases and 48,628 deaths associated with strokes [1]. To distinguish ICH from AIS, hospital-based neuroimaging is still the gold standard, but it is time-consuming since stroke patients must be transferred to stroke centers [2]. In ultra-acute stroke care, time plays a crucial role, as shown by multiple studies proving that treatment can begin within minutes of stroke onset, leading to the “time is brain” proclamation [3,4].

Providing rapid diagnostic procedures upon hospital arrival and reducing treatment duration would certainly improve stroke outcomes. The use of prehospital stroke diagnostic scales has been extensively investigated. A number of scales have been evaluated for predicting large vessel occlusions (LVO) and determining whether thrombectomy is necessary, as well as a clinical prehospital tool for detecting subarachnoid haemorrhage (SAH) and ICH [5,6,7,8]. Machine-learning models have recently been explored as an approach to predict stroke subtypes in the prehospital setting. A model was developed to determine the need for surgical intervention among SAH and ICH patients [9]. These scales and models incorporate a variety of independent variables, including clinical features (such as headaches, aphasia, slurred speech, and weakness of the extremities), physiological parameters (systolic blood pressure (SBP), diastolic blood pressure (DBP), and heart rate (HR)), baseline characteristics, and past medical history (age, hypertension, diabetes mellitus, atrial fibrillation, and renal failure). By integrating cerebral haemodynamic measurements into the predictive framework, we can further refine our understanding of stroke incidents and enhance the predictive accuracy of our models. There is a paucity of data on the integration of cerebral haemodynamic measurements, such as cerebral autoregulation (CA) variables, into stroke-prediction models.

CA is regarded as a vascular self-regulatory process that aims to preserve a constant cerebral blood velocity (CBv) despite variation in arterial blood pressure (ABP), which serves as a protective mechanism for the brain [10]. In cases, where ABP changes suddenly, the transient response of CBv is called dynamic CA (dCA) [10]. The current literature suggests that impaired dCA parameters are associated with worse clinical outcomes and stroke severity [11]. A recent large meta-analysis found that the dCA of the AH predicts clinical outcomes at early stages of AIS, highlighting the potential for using dCA measurements in stroke management in the early stages [12]. Many tools can be used to examine CA, including transcranial Doppler ultrasonography (TCD), which measures CBv in the major cerebral arteries non-invasively.

Multiple clinical studies have demonstrated the effectiveness and feasibility of using a TCD in the ultra-acute setting [13,14,15]. Using TCD in the diagnostic assessment of LVO patients, a systematic review found that reduced flow and asymmetry measurements were predictive of LVO, with sensitivity ranging from 68–100% and specificity ranging from 78–99% [15]. Upon further observational study, TCD improved outcomes and mortality in stroke patients undergoing thrombolytic treatment during the hyperacute phase [14]. In terms of utilising TCD in the prehospital setting, a preliminary study determined its feasibility and effectiveness in traumatic brain injuries and showed that TCD can detect alternation in CBv and be used as therapeutic guidance in the prehospital setting [13]. 

Prior studies have indicated that several peripheral physiological parameters, such as ABP and carbon dioxide (CO_2_) levels, may change after a stroke [16,17]. Two studies described the association between elevated ABP and neurological deterioration among stroke subtypes in the acute phase of stroke care [18,19]. A further retrospective study demonstrated the association between ABP parameters, including BP variability and AIS subtypes in the subacute phase [20]. According to a recent review, hypocapnia was associated with poor prognosis and an increased risk of ischaemic lesions among patients with ICH in the context of lowering BP. Hypocapnia is a key mediator of cerebral ischaemic causing decreased CBF and widening the plateau on the autoregulatory curve [21]. In addition, a comprehensive review found that decreased dCA is related to severe AIS, as well as poorer clinical outcomes and an increased infarct size [11]. Therefore, a close monitoring of EtCO_2_ and dCA status during early phases to detect these alterations may advance stroke therapeutic strategies for either preserving the ischaemic penumbra or lowering BP in ICH.

This comparison study has the potential to provide valuable insights into physiological differences between stroke subtypes during the ultra-acute phase based on physiological observations. The purpose of this preliminary study was to explore the potential of multiple parameters, including peripheral haemodynamic, dCA, baroreceptor sensitivity (BRS), and TFA parameters for characterising each stroke sub-type [22,23].

## 2. Methods

### 2.1. Subjects and Measurements

In this study, 80 patients suffering acute stroke were included whose diagnosis was confirmed by computed tomography within 48 h of the onset of symptoms. Data were retrospectively extracted from the Leicester Cerebral Haemodynamic Database [24]. There was ethical approval for each of the contributing studies (approval from the Research Ethics Committee (REC) for patient studies), and both AIS and ICH recordings were obtained within a median 20 h within a window of 1 to 48 h, with provided written informed consent from participants. Participants were ≥18 years and admitted to the University Hospitals of Leicester NHS Trust (UHL) with mild stroke (based on the National Institute of Health Stroke Scale [NIHSS] < 8). Bilateral simultaneous evaluation of the middle cerebral arteries velocity (MCAv) was performed acutely using TCD with 2 MHz probes (Viasys Companion III; Viasys Healthcare), which were secured using a head frame [25]. Additionally, beat-to-beat ABP was continuously recorded using the Finometer device (FMS, Finapres Measurement Systems, Arnhem, The Netherlands), which was placed on the middle finger of the non-hemiparetic hand of the patient [26]. Breath-by breath end-tidal CO_2_ (EtCO_2_) was determined using a capnograph (Capnocheck Plus) and a nasal cannula (Salter Labs). HR was measured using a 3-lead electrocardiogram (ECG).

### 2.2. Data Analysis

Measurements were recorded continuously using the PHYSIDAS data-acquisition system (Department of Medical Physics, UHL) at a rate of 500 samples per second. The Finometer calibration mechanism (“physiocal”) was switched on and off prior to recordings. Repeated measurements of brachial ABP with a sphygmomanometer (OMRON) were used to calibrate the Finometer output during analysis. A visual inspection was conducted on all measurements with a duration of at least 5 min. Linear interpolation was used to remove narrow spikes (<100 ms), and the CBv recording was then processed via a median filter to remove the remaining spikes. The records were then low-pass filtered with an eight-order Butterworth filter of zero phase and cut-off frequency of 20 Hz. The QRS interval of the ECG was detected automatically to detect the beginning and end of each cardiac cycle. Beat-to-beat HR sequence was visually assessed and manually corrected in the event of missed marks. EtCO_2_ was determined at the end of expiration, linearly interpolated, and resampled with each cardiac cycle.

dCA was estimated using TFA utilising spontaneous fluctuations in ABP as stimulus and CBv as response. TFA has become the method of choice for assessing dCA at rest in clinical and physiological studies [27,28,29]. The main parameters derived from TFA in the frequency domain are the coherence function and gain- and phase-frequency responses. The coherence function ranges between 0 and 1 at each frequency. Coherence refers to the proportion of output power that can be explained linearly by input power [30]. Gain and phase estimation can be used to detect dCA only at frequencies where coherence is above 95% confidence limit according to Claassen et al. [28]. For a recording of 5 min and segments with 512 samples, Fast Fourier Transform (FFT) algorithms are used to derive spectral estimates in the frequency domain, in combination with auto- and cross-spectra, to estimate the coherence function, amplitude (gain), and phase as a function of frequency [28]. The inverse FFT was used to determine the impulse response that was integrated to generate the CBv step response to a sudden alteration in ABP. Autoregulation Index (ARI) was determined by fitting one of the 10 template curves proposed by Tiecks et al. to the CBv step response [31,32].

BRS was estimated through applying spectral analysis of spontaneous oscillation of the SBP and pulse interval (PI) [33,34]. As usually adopted in the literature, the SBP and PI spectra are considered only for the low-frequency (LF) (0.04 to 0.15 Hz) and high-frequency (HF) bands (0.15 to 0.4 Hz). Based on the TFA method, BRS is determined as the average transfer function modulus (gain) of the two previous bands [33,34].

### 2.3. Statistical Analysis

All the parameters were expressed by mean values and standard deviation (SD). Initially, a histogram was used to represent each variable to test the normality of the distribution of the parameters and for identification of outliers. Then, the differences between AIS and ICH for each parameter were tested using parametric independent T-test following assessment of data normality. Statistics were determined to be significant at a *p*-value < 0.05. Statistical analyses were performed using GraphPad Prism 9.0 (GraphPad Software, San Diego, CA, USA).

## 3. Results

A total of 68 patients with AIS [41 male, mean age: 66.82 (SD 12.4) years] and 12 patients with ICH [8 male, mean age: 67.88 (SD 16.23) years] were included. Clinical characteristics and demographic differences between AIS and ICH patients are provided in Table 1, with the cohort demonstrating a mean NIHSS of 6.2 ± 4.6 and 3.8 ± 3.1, respectively. According to the Oxfordshire Community Stroke Project (OCSP) classification, there were nine total anterior circulations (TACS), 33 partial anterior circulations (PACS), 31 lacunars (LACS), and seven posterior circulation syndrome strokes (POCS) [35]. A total of 67 parameters were derived in this study, which related to peripheral haemodynamic, as well as dCA, TFA, and BRS parameters (Appendix A).

### 3.1. Peripheral Hemodynamic and Baroreceptor Sensitivity Parameters

Main peripheral physiological variables were ABP, SBP, DBP, HR, and EtCO_2_, which did not show significant differences between ICH and AIS (Table 1).

The mean BRS gain in the LF range was significantly higher in AIS than in ICH (5.89 ± 5.32 vs. 2.7 ± 1.89 ms/mmHg, *p* = 0.046). Similarly, AIS demonstrated a significantly higher BRS for the LF band compared to ICH (6.7 ± 4.2 vs. 4.10 ± 2.13 ms/mmHg, *p* = 0.04) (Figure 1). Though AIS had a higher value of mean BRS gain in the HF range than ICH, this was not statistically significant (*p* = 0.28). BRS of the HF band and alpha index (average BRSLF + BRSHF) were higher in AIS than in ICH, but the two variables did not differ significantly (*p* = 0.53) and (*p* = 0.22), respectively (Appendix A).

### 3.2. Cerebral Autoregulatory Parameters

dCA variables assessed in both stroke groups included ARI and CBv (systolic velocity, diastolic velocity), in addition to the power of CBv and ABP (Appendix A). ARI had no significant differences between both groups in the affected hemisphere (AH) (ICH, 5.77 ± 2.08 vs. AIS, 5.11 ± 1.87, *p* = 0.286). However, in the unaffected hemisphere (UH), ARI was marginally higher, but the difference was not statistically significant among patients with ICH compared to patients with AIS (5.9 ± 1.7 vs. 4.9 ± 1.8, *p* = 0.072) (Figure 2a). Systolic and diastolic velocities in the AH were significantly higher in ICH than in AIS at 82.57 ± 25.09 vs. 61.94 ± 18.96 cm/s (*p* = 0.002) vs. 32.28 ± 10.007 vs. 25.08 ± 11.82 (*p* = 0.053), respectively. There was no significant difference in the UH for systolic (*p* = 0.18) or diastolic velocities (*p* = 0.31) (Figure 2b).

Furthermore, no significant differences could be observed between AIS and ICH for CBv and ABP power across all frequency bands for both hemispheres (Appendix A). There were a number of different TFA parameters examined in the study for the purpose of comparing stroke subtypes, including coherence function, gain, and phase. There was a significant difference between AIS and ICH for the coherence function in multiple frequency bands in both hemispheres. The coherence function was significantly higher in AIS than ICH in the AH, as determined by very-low frequency (VLF), LF, and HF bands (0.45 ± 0.20 vs. 0.28 ± 0.16, *p* = 0.012), (0.51 ± 0.23 vs. 0.33 ± 0.17, *p* = 0.012), (0.57 ± 0.22 vs. 0.40 ± 0.19, *p* = 0.016), respectively (Figure 3). Similarly, both groups in the UH demonstrated significant differences in the coherence function among LF and HF bands (AIS, 0.51 ± 0.23 vs. ICH, 0.28 ± 0.18. *p* = 0.002) and (AIS, 0.55 ± 0.25 vs. ICH, 0.37 ± 0.23, *p* = 0.02), respectively. However, both hemispheres showed no significant differences in terms of gain and phase parameters between AIS and ICH (Appendix A), though the phase in the LF band showed a significantly higher value for ICH than for AIS in the UH (1.11 ± 0.60 vs. 0.62 ± 0.52 radians, *p* = 0.004.

## 4. Discussion

### 4.1. Main Findings

#### 4.1.1. Differences between AIS and ICH

Multiple differences between patients suffering from ICH and AIS based on parameters extracted from TFA and BRS analyses were identified, including systolic and diastolic velocities of the AH, which were significantly higher with ICH compared to patients with AIS. The coherence function for both hemispheres, in different frequency bands was significantly higher in AIS than in ICH. Similarly, BRS for the LF band and the mean gain of BRS was higher in AIS than in ICH. Variations in these autoregulation variables may have adverse effects through various mechanisms, including myogenic tone, neurogenic responses, metabolic, and endothelial changes [36,37]. CA responses have been attributed to these processes, although they are still controversial as far as their relative contribution and their integration is concerned.

Non-invasive techniques have been used to examine BRS in stroke patients suffering from AIS and ICH. According to our findings, there were significant differences in the main gain of the LF BRS and the absolute value of BRS in the LF band between stroke subtypes. Acute stroke is frequently associated with disruptions of the autonomic nervous system, leading to an increased risk of cardiac arrhythmias and mortality. Previously, BRS impairment has been associated with poor prognosis with stroke and cardiovascular disease [38]. Studies have investigated the relationship between BRS and stroke severity and found that BRS is associated with infarct size in individuals with AIS and haematoma expansion in patients with ICH and poor prognosis [39,40]. The effects of acute stroke subtypes on BRS were assessed in multiple studies and contributed to the understanding of the haemodynamic parameters of stroke patients.

A range of dCA variables has been investigated, including the following TFA variables: phase (which reflects the autoregulatory response), gain (which reflects the damping of dCA), and coherence (which reflects the relationship between frequencies in ABP, as well as CBv, ARI, CBv, and systolic and diastolic velocities). The present study demonstrated a significant difference between the two groups in the diastolic and systolic velocities of the AH and the ARI of the UH was borderline, with a subtle but not significant difference. In addition, the coherence function showed a significantly higher value in the AIS than in the ICH in the AH among all frequency bands. Recently, several studies have examined the impact of stroke (ICH and AIS) on CA. It was demonstrated that CA is impaired following stroke and associated with severity and poor functional outcomes [11,41,42,43,44]. In stroke management, it is important to understand and address CA impairment. Developing interventions and therapies aimed at preserving CA may improve the outcome of stroke patients.

Clinical implications of impaired CA and BRS following a stroke are widely documented. Existing literature indicates the feasibility of using BRS and CA variables to determine the prognostic value, including stroke severity and functional outcomes among stroke patients [45,46].

A recent review of the existing literature on LVO detection techniques was published recently [47]. CA parameters such as CBv using TCD were used in three studies to categorize stroke subtypes [48,49,50]. A study conducted by Thorpe and colleagues assessed the CBv waveforms using TCD between groups with LVO and those without LVO by using unsupervised machine-learning methods [48]. In addition to helping to differentiate between stroke subtypes, the method could be helpful in providing prehospital stroke care as well. A further study investigated the feasibility of detecting LVO using two metrics (velocity asymmetry index and velocity curvature index) based on the differences between velocities across cerebral hemispheres [49]. Compared to the other metrics, the velocity curvature index detects LVO more accurately with a sensitivity of 91% and a specificity of 88% in a separate study [50]. The study indicates that both metrics yield robust information about intracranial occlusion. In terms of the development of LVO prehospital diagnostics based on TCD, both are objectives and real-time computable metrics.

However, there has been little reported evidence of their ability in discriminating between AIS and ICH. Stroke management, prevention, and rehabilitation are the major components of clinical stroke care but do not routinely include BRS and CA parameters as intrinsic components, although these physiological parameters are extensively used in clinical research studies to obtain information regarding the roles of autoregulation and the autonomic nervous system. With regard to the valuable information provided by BRS and CA variables in monitoring and managing stroke patients, combining these variables with other clinical assessments and imaging techniques of stroke subtype identification may facilitate a more comprehensive and proactive stroke care approach.

#### 4.1.2. Methodological Considerations

As in both AIS and ICH, elevated blood pressure is most common in the acute phase and is associated with early neurological deterioration in ICH patients [18,19]. Despite its prognostic value and critical role in the management of acute stroke patients, in our study, neither SBP nor DBP indicated significant differences between stroke subtypes (Table 1), thus suggesting that a more detailed analysis of its variability in the ultra-acute phase should receive more attention in future studies aiming at the early classification of AIS and ICH. Monitoring blood CO_2_ levels during acute stroke management is essential since its elevation acts as an effective vasodilator and contributes to substantial increases in CBv [51]. Recently, a meta-analysis of 20 studies and 660 acute stroke patients examined CO_2_ levels in acute stroke patients [17]. The study reported that acute stroke patients are significantly more likely to be hypocapnic, providing evidence that routine CO_2_ measurements are beneficial in acute stroke patients. In our study, no significant differences were found between AIS and ICH based on EtCO_2_ in the absence of any intervention. A recent prospective study evaluated the feasibility of CA-targeted intervention among ICH patients using a hyperventilatory maneuver [51]. Minhas et al. demonstrated an improvement in CA secondary to hyperventilation-induced hypocapnia [51]. A further study is required to investigate the effectiveness of inducing hypocapnia through hyperventilation in more severe stroke cohorts.

With regards to the dCA parameters, a prospective study included 26 ICH patients and examined the dCA (gain and phase) conditions during the acute stage and how dCA may be associated with clinical outcome [45]. Qeinck et al. indicated that the phase did not differ between ICH and control, nor did it differ between hemispheres. However, gain was impaired in acute ICH, but it was not correlated with any clinical outcomes. Furthermore, a prospective study, which included 34 ICH patients, examined the association of dCA variables with the clinical characteristics and outcomes [43]. The study reported that the average phase did not significantly differ between the AH and UH among ICH patients, but a significantly lower phase was found in both hemispheres, which indicated impaired dCA and was associated with a worse 90-day modified Rankin Scale and poorer Glasgow coma scale score. Although gain did not significantly change between both hemispheres in ICH, a higher gain was significantly associated with larger haematoma volume and higher NIHSS. Interestingly, no previous study has compared CA parameters between stroke subtypes, as in our study, where bilateral CA parameters were evaluated between AIS and ICH. There were no significant differences in phase and gain between AIS and ICH in either hemisphere with the exception of phase in the LF band in the UH which showed higher values in the ICH group.

In light of exploring the haemodynamic characteristics of severe stroke patients and their potential implications, a couple of prospective studies have examined the effect of stroke severity on cerebral hemodynamic parameters.

According to Salinet et al., a higher CBv in both hemispheres and an impaired CA were associated with stroke severity and worse outcomes 3 months after the stroke [52]. Similarly, Llwyd et al., found that higher HR and CBv of the UH were associated with stroke severity. However, no significant differences in ARI were found [53]. 

#### 4.1.3. Clinical Perspectives

In prehospital stroke care, the main objective is to ensure that patients are stabilised and transported, as fast as possible, to an appropriate facility. Several prehospital stroke scales and scoring systems were examined in a prospective study and found to be capable of detecting strokes in the prehospital setting with a reasonable level of sensitivity and specificity [6]. Throughout stroke care, the impact of physiological parameters on acute stroke outcomes has been increasingly considered.

A number of physiological parameters have been shown to be associated with worsened neurologic conditions and an increased mortality rate [54,55,56]. As a result, the question has arisen as to whether increased prehospital physiological parameter monitoring would be beneficial in identifying stroke subtypes and enabling earlier intervention. The results of a previous study revealed physiological differences between stroke and stroke mimics in prehospital care (such as glucose levels, ABP, body temperature, oxygen saturation, and HR with continuous measures) [57]. Utilising physiological variables monitoring to discriminate stroke subtypes could result in a novel approach to acute stroke care, leading to improvements in its effectiveness. Introducing prehospital monitoring involves practical obstacles, including cost, equipment availability, personnel training, and the ability to transmit data onto hospitals. Further work will be needed to identify confounding factors and to establish wider-ranging models that might improve the ability of multiple parameters to discriminate between AIS and ICH. In ultra-acute stroke patients, measuring dCA and autonomic dysfunction might provide valuable diagnostic or prognostic insight, allowing for better care to be delivered. The classification of stroke subtypes at an earlier time point (4 h after symptoms onset) requires investigation in future prospective studies.

### 4.2. Limitations of the Study

There are multiple limitations in our study: the estimation of cerebral blood flow based on CBv measurements in the MCA is only valid if the diameter of the MCA remains constant. CARNet White Paper proposes consensus guidelines protocol for dCA research in order to address this limitation by introducing several recommendations for data collection (environmental status for rested and positioned appropriately during measurements), physiological measurements and equipment utilised, and acceptable recording duration, which enhances the reliability of cerebral blood flow estimates [28]. As in our study, this is the case when participants are at rest. Despite the fact that all data were collected using a standard protocol and in the same research facility, in this retrospective study, data were collected by different TCD personnel, which may result in a wider range of CBv values and other parameters. It is important to note that due to the difficulties confirming pre-existing diagnoses at the time of admission, data on risk factors were incomplete. However, with the lack of relevance to our approach, these missing data do not affect our models. The sample size for this study was 80 participants, who were collected retrospectively. Further prospective studies with a broader mixed-sex cohort are necessary to support the current findings. Lastly, it is essential to note that the results of this study are limited to the use of TFA within the set of parameters recommended by existing guidelines [28]. Consequently, ABP-CBv dynamic relationship models could be further extended using time-domain techniques as proposed by Panerai et al., Czosnyka et al., and Nogueira et al., or closed-loop models as proposed by Marmarelis et al. [58].

The absence of more significant parameters could have been due to a number of factors. Our sample size was relatively small, based only on 80 participants retrospectively collected. Larger sample sizes will also allow for more in-depth analysis of those variables that showed significant differences between the two types of strokes (Appendix A), identification of co-factors, and more elaborate multivariable predictive models. In particular, identifying differences in parameter behaviours due to sex differences is of paramount importance and should also benefit from larger samples. Above all, the main ambition of this line of work is to anticipate a future when a more comprehensive set of physiological measurements will be available to paramedics to anticipate stabilization funding and care of stroke patients, according to evolving guidelines. 

## 5. Conclusions

In conclusion, the current study revealed several differences between patients suffering from ICH and AIS based on parameters extracted from transfer function and baroreceptor sensitivity analyses, highlighting the need for further multivariate statistical modeling to improve the ability of multiple parameters to differentiate between AIS and ICH quantitatively. Furthermore, prospective studies investigating the classification of stroke sub-types at an early stage are warranted.

## Figures and Tables

**Figure 1 healthcare-12-00966-f001:**
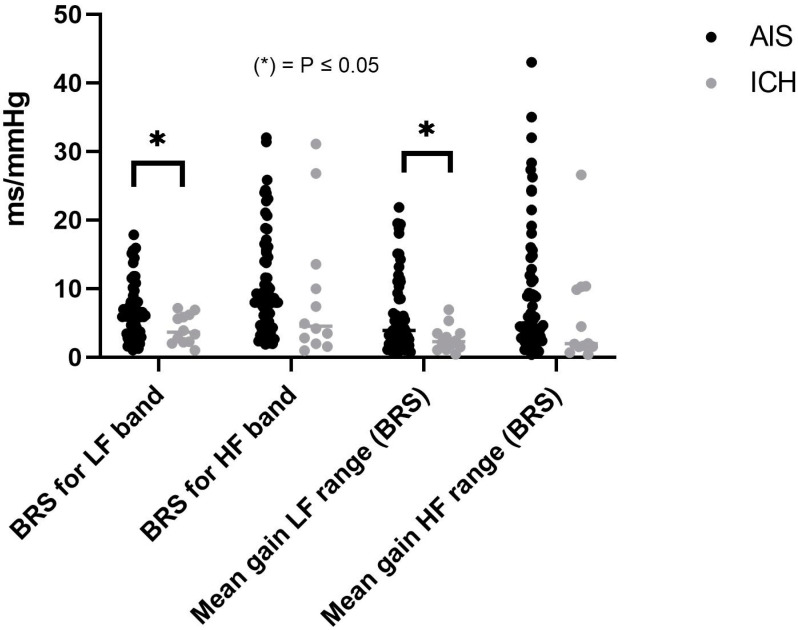
Baroreceptor sensitivity (BRS) in different frequency bands.

**Figure 2 healthcare-12-00966-f002:**
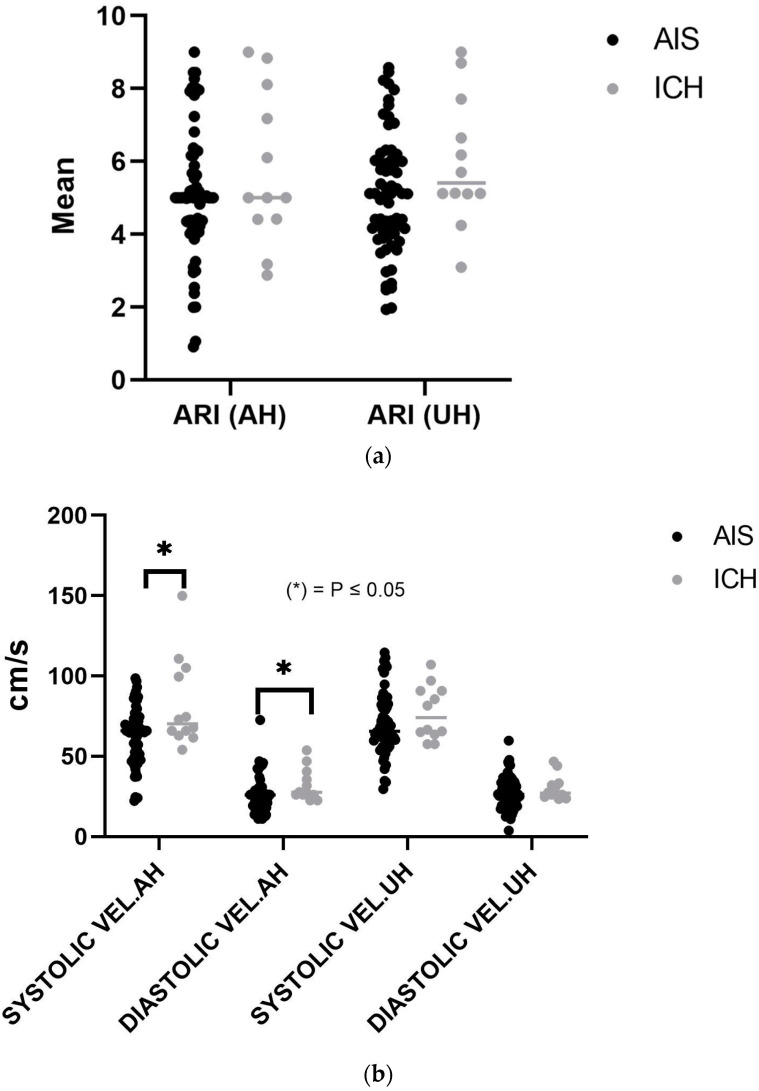
(**a**) Autoregulation index (ARI) for both hemispheres. (**b**) Bilateral systolic and diastolic velocities.

**Figure 3 healthcare-12-00966-f003:**
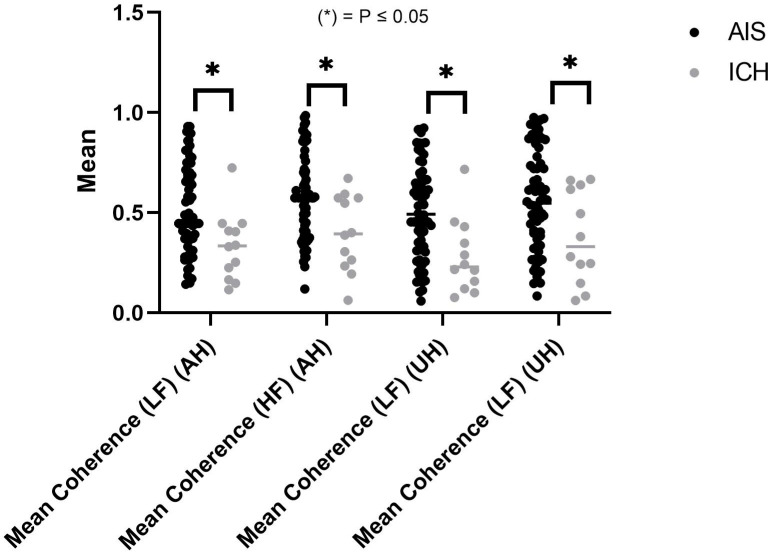
Coherence function for both hemispheres in different frequency bands.

**Table 1 healthcare-12-00966-t001:** Demographic and clinical characteristics of the patients with two different stroke subtypes.

Characteristic	AIS *n* = 68	ICH *n* = 12	*p*-Value
Age, years, mean (SD)	66 (12)	68 (16)	0.80
Sex (male), *n* (%)	41 (60.3)	8 (66.7)	0.67
NIHSS admission, median (IQR) (SD)	6 (4)	3.5 (3.5)	
Time to Assessment, hours (SD)	18.64 (13.5)	24.08 (11.1)	0.19
OCSP (Stroke subtype), *n* (%)			
TACS	8 (11.8)	1 (8.3)	0.72
PACS	29 (42.6)	4 (33.3)	0.54
LACS	27 (39.7)	4 (33.3)	0.67
POCS	4 (5.9)	3 (25)	0.03
Hemodynamics parameters			
Systolic BP, mean (SD)	148.3 (27.1)	145.2 (24.6)	0.69
Diastolic BP, mean (SD)	81.7 (15.2)	75.3 (15.3)	0.19
End-tidal CO_2_, mean (SD)	33.4 (3.0)	34.9 (4.0)	0.43
Heart Rate, bpm	70.1 (13.6)	72.6 (13.6)	0.60
ABP, mean (SD), mmHg	102.6 (17.5)	99.2 (15.05)	0.48

NIHSS, National Institutes of Health Stroke Scale; AIS, Acute ischaemic stroke; ICH, Intracerebral haemorrhage; OCSP, Oxfordshire Community Stroke Project; TACS, Total anterior circulation stroke; PACS, partial anterior circulation stroke; LACS, lacunar stroke syndrome; POCS, posterior circulation syndrome; BP, Blood Pressure; CO_2_, Carbon Dioxide; ABP, Arterial Blood Pressure.

## Data Availability

The original contributions presented in the study are included in the article/Appendix A, further inquiries can be directed to the corresponding author/s.

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
