# Peer review of "A Multi-Parametric Approach for Characterising Cerebral Haemodynamics in Acute Ischaemic and Haemorrhagic Stroke"

_healthcare, 2024, doi:10.3390/healthcare12100966_

Round 1

Reviewer 1 Report

Comments and Suggestions for Authors

Reviewer 2 Report

Comments and Suggestions for Authors

Authors of “A Multi-Parametric Approach for Characterising Cerebral Haemodynamics in Acute Ischaemic and Haemorrhagic Stroke” in presented article aimed to explore the potential of multiple haemodynamic parameters in characterising stroke sub-type. At present, due to lack of specific clinical symptoms of ischemia ora bleeding, brain imaging techniques remain the gold standard for differentiation between ischaemic and haemorrhagic stroke. There are efforts to design scales that would help to diagnose the type of stroke in prehospital settings, however their sensitivity and efficacy leave much to be desired. Incorporation of objective measurement of haemodynamic parameters could possibly help in both prehospital assessment as well as further diagnostic and treatment decisions in stroke patients. 

In this retrospective study 80 acute stroke patients were included. The diagnosis was confirmed by computed tomography within 48 hours of the onset of symptoms. Data were retrospectively extracted. Participants were all adults, with mild stroke (NIHSS <8).  Evaluated parameters included: bilateral simultaneous evaluation of the middle cerebral arteries velocity (MCAv) using transcranial Doppler(TCD), continuously recorded beat-to beat ABP,, breath-by breath end-tidal CO2 (EtCO2), HR measured using a 3-lead electrocardiogram (ECG). Dynamic cerebral autoregulation (dCA) was estimated using TFA utilising spontaneous fluctuations in ABP as stimulus and cerebral blood velocity (CBv) as response.

Main peripheral physiological variables (ABP, SBP, DBP, HR and EtCO2 ) did not show significant differences between ICH and AIS. aCA assessment demonstrated some significant differences between ICH and AIS regarding auto regulation index (ARI) in unaffected hemisphere as well as systolic and diastolic velocities in the affected hemisphere which were significantly higher in ICH patients.

The study is interesting and novel, however at this stage it appears to bring little impact to clinical practice, especially in the prehospital level. The article is properly structured, the language is comprehensive and supplementary material is graphically clear. However I need to address some substantive problems that in my opinion, if properly addressed, could significantly improve the soundness of proposed article.  

  In the introduction authors refer to prehospital diagnostic scales used for differentiation of stroke types and then only vaguely mention that haemodynamic parameter assessment can be used to improve the diagnostics and treatment in the course of stroke. It would be advised to describe and explain how exactly these parameters can be incorporated in diagnostic and therapeutic algorithms especially when timing is limited as it is in emergency stroke care. 

The other matter would be the technical issues associated with these measurements. As authors presented in the limitations section - eg. TCD can provide a valid estimation of cerebral blood flow only valid if the diameter of the MCA remains constant (the patient is at rest), also the measurements can be confounded by comorbidity or patient characteristics. Can authors propose any solutions regarding how these limitations can be approached and avoided in everyday practice?

At last patients included in the assessment were suffering mild stroke (NIHSS <8). In these cases the size of the lesion would be usually small and could have limited impact on haemodynamics. How do authors reason choosing this group of patients for their assessments. It would be interesting to address the issue of haemodynamics in severe stroke and include this in the discussion. 

I would be happy to reassess the article after authors refer to all the issues mentioned above. 

Reviewer 3 Report

Comments and Suggestions for Authors

I would like to congratulate you on your current article. It addresses a particularly important issue, and you have presented it comprehensively.
Please let me present some of my thoughts that could probably help you:

 -       Clarification of Stroke Impact: Consider providing more specific statistics or references to quantify the impact of stroke globally and in the UK. This can enhance the introduction's effectiveness in highlighting the urgency and significance of the research topic.
-       Transition Enhancement: To improve the flow between paragraphs, consider adding a transitional sentence or phrase at the end of the paragraph discussing machine-learning models (lines 52-60) to smoothly introduce the discussion on cerebral hemodynamic measurements in stroke prediction models.
-       Explanation of Cerebral Autoregulation (CA): While the introduction briefly introduces CA, offering a concise explanation of its significance and role in stroke management could enhance clarity, especially for readers less familiar with the concept. This could help to strengthen the rationale for including CA measurements in stroke prediction model
-       Equipment Description: Providing a brief explanation or reference for the equipment used, such as TCD (Transcranial Doppler) with 2MHz probes and Finometer device for continuous ABP recording, could help readers unfamiliar with these technologies better understand the methodology.

Round 2

Reviewer 2 Report

Comments and Suggestions for Authors

Just a note that in part of paragraph 4.2.1 mentioned papers ( Salinet et al., Lloyd et al.) are not included in the references. That needs to be updated. Other than this I am happy with provided corrections. 

Author Response

I appreciate your comments regarding adding references. In the revised manuscript, the necessary references have been added to lines 327 and 330 of the discussion.